# Quality Evaluation and Characterization of Specific Spoilage Organisms of Spanish Mackerel by High-Throughput Sequencing during 0 °C Cold Chain Logistics

**DOI:** 10.3390/foods9030312

**Published:** 2020-03-09

**Authors:** Ruihang Zheng, Xiaorong Xu, Jiali Xing, Hai Cheng, Shufen Zhang, Jian Shen, Hesheng Li

**Affiliations:** 1Ningbo Institute for Food control, Ningbo 315048, China; ruixing161@gmail.com (R.Z.); 1711091100@mail.nbu.edu.cn (X.X.); 1811085028@mail.nbu.edu.cn (H.C.); p1809085214@cjlu.edu.cn (S.Z.); 1711091085@mail.nbu.edu.cn (J.S.); 2College of food and pharmaceutical sciences, Ningbo University, Ningbo 315211, China; lihesheng@nbu.edu.cn

**Keywords:** Spanish mackerel, cold chain logistics, quality, specific spoilage organisms, high-throughput sequence

## Abstract

Exploring the spoilage mechanism of Spanish mackerel is important to reduce the waste of Spanish mackerel and extend its shelf life. Cold chain logistics are commonly used to maintain the high quality and prolong the shelf life of aquatic products in circulation and storage. We assessed the sensory (body surface, odor, fish gills, fish elasticity, eyes, and overall assessment), chemical (total volatile base nitrogen (TVB-N), pH and 2-thiobarbituric acid (TBA)), and microbial characteristics (total viable counts (TVCs) and lactic acid bacteria) of Spanish mackerel combined with high-throughput sequencing at frequent intervals to determine their freshness and specific spoilage organisms (SSOs) during 0 °C cold chain logistics. Results showed that TVB-N, TBA, and TVCs correlated well (R^2^ > 0.90) with the sensory scores with prolonged circulation and storage time. The SSOs of Spanish mackerel were Proteobacteria in phylum and Pseudomonas in genus. The shelf life of mackerel under the 0 °C ice-stored cold chain system was approximately seven days, which is roughly three days longer compared with the traditional low-temperature storage method. These findings indicated that the freshness evaluation of Spanish mackerel in cold-chain circulation could be achieved by selecting appropriate chemical, microbial, and sensory indices. The study contributes to extend the shelf life of cold-chain Spanish mackerel by inhibiting the growth of dominant bacteria and provides a basis for the development of methods and tools to predict the shelf life of Spanish mackerel.

## 1. Introduction

Spanish mackerel (Scomberomorous niphonius), belonging to family Scombridae (Percida), is the most commercially important pelagic species in the northern Persian Gulf [1,2]. Besides, Spanish mackerel is one of the most important aquaculture species in China, especially in east and southeast China [3]. After the large and small yellow croaker, cuttlefish, and hairtial resources sharply decreased, Spanish mackerel has become the main economic fish in China in recent years given its wide distribution, fast growth, and high yield [4]. Mackerel has become an indispensable delicacy in coastal areas because of its delicious taste and extremely high nutritional value (high protein and unsaturated fatty acid content) [5]. However, the survival of Spanish mackerel, like most marine fish, after landing is difficult, so maintaining its high quality during storage and transportation is challenging [6]. If the factors affecting its quality are not controlled during its circulation and storage, then unnecessary economic losses and waste of resources are bound to ensue. Studying the deterioration mechanism of Spanish mackerel during cold-chain logistics is important to prolong its shelf life and maintain high quality.

Many factors contribute to the spoilage of Spanish mackerel. The degradation of the abundant proteins and fats of Spanish mackerel produces small molecules of alkaline materials, such as biogenic amine and nitrogen compounds [7] and sulfur-containing compounds during storage and transportation [8]. These substances are the main cause of the stench and deterioration of Spanish mackerel [9,10,11]. The compounds can serve as easily digestible nutrients for microbial growth [12], which further aggravates the spoilage of Spanish mackerel. However, only a few members of the microbial community called specific spoilage organisms (SSOs) for freshly caught or processed seafood contribute to seafood spoilage [13]. Finding the SSOs and the effective logistics or storage methods for Spanish mackerel is necessary to slow down the deterioration rate of mackerel. Several techniques, such as real-time polymerase chain reaction (PCR), denaturing gradient gel electrophoresis, or temporal temperature gradient gel electrophoresis and terminal restriction fragment length polymorphism have been used to study spoilage organisms. However, these techniques cannot fully examine the microbial species and their proportions in specific samples. Hence, the emerging high-throughput sequencing (HTS) technology has attracted much attention for its satisfactory efficiency and accuracy in microbiota analyses [14]. HTS has been successfully used to characterize the SSOs of various seafoods such as cod and salmon fillets, cold-smoked salmon, oysters, and raw steaks of yellowfin tuna [15,16,17]. Researching the SSOs of Spanish mackerel by HTS is valuable.

Cold-chain logistics are commonly used to maintain the high quality and long shelf life of agricultural products in long-distance circulation and storage [18,19,20]. Several researchers have assessed the effects of cold-chain logistics on the quality maintenance of grape [21], hairtail fish, and shrimp under certain conditions [22,23]. Results from these studies proved that cold-chain logistics are more efficient than conventional storage for the preservation of agricultural products. However, studies on Spanish mackerel are limited to the effects of different storage temperatures on its quality [24]. Therefore, we investigated the deterioration rate of Spanish mackerel during 0 °C cold-chain logistics to explore its effectiveness in maintaining the good quality of Spanish mackerel. In addition, we identified the SSOs that lead to the spoilage of Spanish mackerel in cold-chain logistics. SSOs produce various volatile compounds, some of which can function as indicators of spoilage. Total volatile basic nitrogen (TVB-N) is a volatile amine widely used as a parameter to reveal microbiological spoilage of seafood [25,26].

In this context, microbial (total viable count (TVC) and lactic acid bacteria (LAB)), chemical (TVB-N, 2-thiobarbituric acid (TBA), and pH), and sensory activities (body surface, odor, fish gills, fish elasticity, eyes, and overall assessment) of Spanish mackerel in circulation and storage time (216 h) during 0 °C cold chain logistics were explored and compared with the traditional low-temperature storage method to examine the effect of cold-chain logistics. Furthermore, HTS was used to demonstrate the bacterial diversity and composition of the SSOs of the cold-chain Spanish mackerel in order to find the spoilage mechanism.

## 2. Materials and Methods

### 2.1. Samples Preparation and Storage Conditions

Freshly caught Spanish mackerel (*S. niphonius*) were collected from the Zhoushan fishing ground, the most important Spanish mackerel fishing area in China, in November 2018 and transported to the laboratory in ice within 30 min. The mean weight and length of fish were roughly 510 ± 3 g and 40 ± 2 cm, respectively.

Whole fish were stored in a cooler box by alternating a layer of fish placed on the belly with a layer of flake ice and kept in a cold room (4 °C). Melting water was drained and ice was replaced when necessary to maintain the temperature of these samples at 0 °C. Temperature variation was simulated based on the method of Yang et al. [22]. Temperature was monitored through a digital thermometer (Fluke-NetDAQ32 Multipoint temperature collector, T type and variability) by placing the probes inside the center back of Spanish mackerel [27] in the box and then recording the temperature of the samples, box, and room [28]. Three samples of fish were randomly selected at each time for index detection. The sample sites were consistent each time, and the middle part of the fish back was uniformly collected for the experiment. Sterile Spanish mackerel fillets were obtained by the method described by Mace et al. [29] with some modifications. Specifically, ice was made from sterile water. The cooler and the implement used for handling the sterile Spanish mackerel fillets were all sanitized by chlorine solution and 75% ethyl alcohol.

### 2.2. Cold-Chain Simulation Test Design

Fresh Spanish mackerel was selected to simulate the cold-chain flow (Figure 1). From the beginning of the experiment to the end of the simulated cold-chain circulation and distribution, foam boxes of fresh Spanish mackerel were placed in 4 °C, and terminal sales were carried out in 4 °C refrigerated display cabinets with ice.

### 2.3. Microbiological Analysis

Whole Spanish mackerel samples were aseptically cut into small pieces. Afterwards, a 25 g sample was transferred aseptically to a sterile stomacher bag and diluted 10 times in physiological saline peptone solution (0.85% NaCl and 0.1% peptone). The mixture was homogenized for 60 s using a stomacher to obtain the first dilution from which successive decimal dilutions were prepared. TVCs were incubated at 30 °C for 48 h and measured as aerobic plate counts [30,31,32]. LABs were enumerated on double-layered plates of de Man, Rogosa, and Sharp agar (MRSA) incubated at 30 °C for 72 h [33].

### 2.4. Chemical Analysis

TVB-N (in milligrams per 100 g of Spanish mackerel) was measured using the method recommended by Sylvain et al. [34]. In a typical procedure, Spanish mackerel (5 g) was homogenized with 45 mL of 6% perchloric acid (Sigma-aldrich Trading Co., Ltd., Shanghai, China) solution for 2 min. Extract (25 mL) was filtered, alkalinized with 20% sodium hydroxide and subjected to steam distillation. The distillation apparatus was set to produce approximately 50 mL of distillate in 5 min. Distillation ended after exactly 5 min, allowing the same distillation rate for all samples. The volatile base components absorbed by boric acid solution (3%) in laboratory standard glassware (beaker, flask) were determined by titration using hydrochloric acid solution (0.01 mol).

For pH measurement, Spanish mackerel (10 g) was homogenized with 20 mL of distilled water, and pH was measured with a digital pH meter (PHS-3C digital pH meter, Hongyi Instrumentation Co., Ltd., Shanghai, China).

TBA content was determined by the method of Wang et al. [35] and was expressed in milligrams of malonaldehyde equivalents per kilogram of fish flesh. The homogenized sample (2 g) was placed in a 50 mL centrifuge tube to which TCA (50%, 16 mL) and 100 μL butylated hydroxytoluene were added in an Ultra-Turrax (Model T25, Shanghai Kehuai Instruments Co., Ltd., Shanghai, China), and homogenized (8000 rpm, 2 min). The mixture was filtered through Whatman No.1 filter paper (Model 1001-150, Shanghai Root Biological Technology Co., Ltd., Shanghai, China). The 2-TBA solution (0.01 M, 1mL) and filtrate (5 mL) were mixed. The mixture was heated in a boiling water bath for 40 min, cooled to room temperature, and the absorbance of the resultant colored solution was at 532 nm using a Shimadzu spectrophotometer (Model 1601, Shimadzu, Kyoto, Japan). Finally, TBA values were calculated.

### 2.5. Sensory Analysis

Changes in the sensory characteristics of Spanish mackerel samples were evaluated by an experienced panel comprising of 10 people based on a 10-point scale suggested by Don et al. [36]. Based on preliminary visual and olfactory acuity tests, 10 people from the permanent of Ningbo Institute for Food Control (NIFC) were selected to evaluate the sensory quality of Spanish mackerel. Three training sessions were carried out using fresh and spoiled samples of Spanish mackerel in order to establish a list of descriptors and a fairy uniform degree of sensory evaluation (Table 1). The panelists were asked to assign a score of 1–10 for body surface, odor, fish gills, fish elasticity, and eyes (Table 1). Overall acceptability was calculated by taking the average of all five parameters. Time of sensory rejection was defined as the time when the average score of overall acceptability was below 5.

### 2.6. High-Throughput Sequence

#### 2.6.1. DNA Extraction, Amplification and Sequencing

The total DNA of microorganisms from Spanish mackerel was extracted via a Purelink Microbiome DNA Purification Kit [37] (Thermo Fisher Scientific Co., Ltd., Shanghai, China). Target DNA was amplified by PCR as follows—4 µL of 5 × FastPfu buffer, 2 µL of 2.5 mM deoxynucleotide triphosphate, 0.8 µL of 5 µM forward primer, 0.8 µL of 5 µM reverse primer, 0.4 µL of FastPfu polymerase, 0.2 µL of bovine serum albumin, 10 ng of template DNA, and complemented molecular-grade water to 20 µL. The thermocycler program consisted of a denaturation step of 3 min at 95 °C; followed by 28 cycles of 30 s at 95 °C, 30 s at 55 °C, and 45 s at 72 °C; and a final elongation step of 10 min at 72 °C. Specific primer 338F (5′-ACTCCTACGGGAGGCAGCA-3′) and 806R (5′-GGACTACHVGGGTWTCTAAT-3′) were used to amplify the V3 and V4 regions of the 16s rDNA gene. Tag-encoded HTS was carried out by Illumina MiSeq platform (Shanghai Majorbio Bio-pharm Technology Co., Ltd., Shanghai, China).

#### 2.6.2. Data Processing and Taxonomic Classification

Raw sequencing data were analyzed and refined through the quality clinic process chart provided by Quantitative Insights Into Microbial Ecology software version 1.6.0 [38] to guarantee a higher level of accuracy in terms of operational taxonomic unit (OTU) detection. In brief, the final effective tags were obtained after detection and removal of chimeric sequences [39]. OTUs defined by a 97% similarity were picked using the UPARSE software [40], and representative sequences were submitted to the Ribosomal Database Project’s classifier [41] to obtain the taxonomy assignment on the phylum and genus levels [17].

### 2.7. Statistical Analysis

All of the tests were performed in triplicate. Data are presented as mean ± standard deviation (SD). One-way ANOVA was performed with SPSS (Version 13.0, SPSS Inc., Chicago, IL, USA) and followed by Fisher’s least significant difference, to verify significant differences between samples. Results were considered significant when *p* < 0.05. Pearson’s correlation test was conducted to determine the correlation between variables.

## 3. Results and Discussion

### 3.1. Results of Microbiological Analysis

Many factors can cause the spoilage and deterioration of aquatic products. Microorganisms are the main factors, and TVC is generally used to determine the freshness of aquatic products [42]. The TVC value of freshly harvested aquatic products is usually 3.00–4.00 log CFU/g; hence, the TVC value below 4.00 log CFU/g is generally regarded as fresh fish. Fish has started to degrade when the TVC value exceeds 5.00 log CFU/g [43]. The TVC and LAB values of cold-chain Spanish mackerel are shown in Figure 2. The initial count for TVC was 3.87 ± 0.14 log CFU/g, which was similar to that in the research of Otero et al. [44] in Atlantic mackerel (*Scomber scombrus*) and Sofi et al. [45] in Indian mackerel (*Rastrelliger kanagurta*). The TVC value reached 8.03 ± 0.14 log CFU/g when Spanish mackerel was stored for 216 h. The TVC value increased linearly throughout the storage period and is different from LAB counts, which had a nonlinear trend. The initial counts of LAB were 2.83 ± 0.23 log CFU/g and then increased to 4.27 ± 0.25 log CFU/g at 120 h but decreased to 3.75 ± 0.24 log CFU/g finally.

### 3.2. Results of Chemical Analysis

As good indicators of spoilage, volatile compounds, such as ammonia, trimethylamine, and dimethylamine are the major components of TVB-N [46,47]. The TVB-N of cold-chain Spanish mackerel is shown in Figure 3. The initial TVB-N of Spanish mackerel was 11.43 ± 0.83 mg/100 g, which was similar to those usually reported of fresh mackerel that range between 10 and 20 mg/100 g [48,49]. TVB-N showed an increasing trend during storage with a final value of 31.01 ± 0.58 mg/100 g (Figure 3), which exceeds the upper limit (25 mg/100 g) in the report of Okpala et al. [50]. Increased TVB-N in seafood during cold-schain storage is due to the formation of basic compounds when fish muscle is degraded by enzymatic reactions and microbial activity [51]. The TVB-N of cold-chain Spanish mackerel increased throughout the entire process of circulation and storage with prolonged storage time. The trend of TVB-N in Spanish mackerel was similar to that reported by Wang et al. [52] in cold-chain salmon. Both grew rapidly in the later stage of storage.

Changes in pH could also indicate the freshness of aquatic products to a certain extent. The initial pH of Spanish mackerel was 6.75 ± 0.03, which was similar to the fresh Spanish mackerel (pH = 6.67) [53]. However, the pH decreased to 6.37 ± 0.02 at 24 h and then increased to 6.72 ± 0.01 at the end of storage (Figure 3). This change of pH is due to the rigidity and autolysis of Spanish mackerel after death [53], during which the muscle tissues of fish undergo glycolysis in the production of lactic acid, resulting in the decreased pH of Spanish mackerel. The trend of pH in our research was similar to that reported by Pinter et al. [54] in irradiated mackerel (*S. scombrus*) during storage at 4 °C. Results of LAB (Figure 2) and pH values showed a certain correlation between the number of LAB and pH value. The pH value decreased when the number of LAB increased which means that acid produced by LAB made a certain contribute to the reduction of pH value. Besides, the bacterial degradation of organic compounds in Spanish mackerel supplemented with autolytic activity and its end products may explain the increase in pH of Spanish mackerel [55]. Our results were similar to those reported by Huang et al. [56] in cold-chain red drum (*Sciaenops ocellatus*). Besides, the increase in pH between 0 and 216 h was less than one unit. This slow rise in pH was also observed by Xie Tingting [53]. Bennour et al. [57] have proven that pH is not an efficient parameter to evaluate the quality of mackerel. Therefore, we could speculate that pH was an inappropriate spoilage indicator for cold-chain Spanish mackerel.

TBA content is generally used to reflect the degree of fat oxidation in aquatic products. The principle is that the malondialdehyde produced by the oxidation degradation of unsaturated fatty acids reacts with TBA to form stable red compounds [58]. Changes in TBA content during the circulation and storage of cold chain Spanish mackerel are shown in Figure 3. The TBA content of Spanish mackerel increased during storage. The initial TBA value of mackerel was 0.183 ± 0.065 mg/kg, and the terminal TBA value of cold-chain mackerel was 0.663 ± 0.078 mg/kg, which increased to 1.344 ± 0.041 mg/kg at 216 h. The TBA content of cold-chain mackerel increased faster in the later period of circulation than in the earlier period, which might be related to the direct exposure to air during the sale stage, because oxygen in the air could accelerate the oxidation of unsaturated fatty acids in Spanish mackerel.

### 3.3. Results of Sensory Analysis

The appearance quality of Spanish mackerel directly affects its value and sales volume in the process of sale. Results of the sensory evaluation of Spanish mackerel are presented in Table 2. At the beginning of the storage process, Spanish mackerel did not produce any off odor, and all sensory indicators indicated that the mackerel was fresh. Sensory quality decreased with prolonged storage time and reached the lower limit of the acceptable range starting from 168 h onwards. Mendes et al. [59] suggested that the shelf life of horse mackerel (*Trachurus picturatus*) stored on ice with controlled temperature (3 °C) is between three and four days, which was shorter than that measured in our research. Our results indicated that the cold-chain process is effective for extending the shelf life of mackerel. The length of time for each sensory index to reach the five-point limit decreased as follows—body surface > odor > fish gills > fish elasticity > eyes.

### 3.4. Relationship between Sensory Scores and Microbial Concentration or Chemical Indicators

We studied the relationship between sensory quality and chemical or microbial characteristics to explore the spoilage mechanism of Spanish mackerel. A good correlation (R^2^ > 0.90) was found between TVC, TVB-N, or TBA production and the mean sensory scores by the 10 panelists at each sensory session from the beginning of storage until the sensory rejection time (Figure 4). We could infer that TVC, TVB-N, or TBA, combined with sensory quality, is a good way to judge the freshness of Spanish mackerel. This finding was consistent with the results of TVB-N and TBA changes in Section 3.2. Specifically, both TVB-N and TBA content reached the upper limit when the quality of Spanish mackerel was below the sensory rejection.

### 3.5. Results of OTU Clustering and Species Annotation

We relied on HTS to study the SSOs of Spanish mackerel, which are the major microorganisms that lead to its spoilage, to investigate the decomposition of Spanish mackerel. The 16S rDNA of bacteria is the ideal molecular label for species identification due to its species specificity, appropriate molecular size, and low mutation rate. The average length of the valid target bands of the PCR products was approximately 441–460 bp, which is within the sequence length scope of V3 + V4 zone of 16S rDNA. The samples obtained valid reads within the range of 41,386–62,616 after filtering low-quality reads and trimming adapters, barcodes, and primers OTU numbers ranged from 36 to 243, and initially decreased before increasing (Table 3). This result indicated that microorganisms in cold-chain Spanish mackerel might adapt to the living environment when the storage conditions changed with prolonged storage time.

Microbial richness estimators (ACE and Chao1) and diversity indices (Simpson and Shannon) were used to evaluate the alpha diversity within a certain Spanish mackerel sample (Table 4). ACE and Chao1 are both indices that estimate the OTU numbers in a microbial community but are calculated according to different algorithms. Simpson and Shannon indices are used in ecology studies to quantitatively decide the biodiversity in a region. Simpson index is negatively correlated with biodiversity, whereas Shannon index is positively related to biodiversity.

Among the bacteria, M168 had the highest richness and diversity (Shannon: 2.47, Simpson: 0.13, ACE: 328.33, Chao1: 182.10), whereas M72 had the lowest bacterial richness and diversity (Shannon: 1.92, Simpson: 0.29, ACE: 109.21, Chao1: 84.11). The sample’s coverage is an estimator of sampling completeness, which calculates the probability that a randomly selected amplicon sequence from a sample has been already sequenced. A good coverage indicates an adequate level of sequencing to identify the majority of diversity in the samples. Table 3 shows that the values of estimated sample coverage were > 0.99. This value meant that the information was sufficient to reveal the microbiota of samples.

Relative abundance of bacterial community proportions in phylum (A) and genus (B) levels are shown in Figure 5. Four major phyla were found, and Proteobacteria (64.06%–82.50%) was the dominant phylum in all the samples, followed by Firmicutes (6.05%–31.8%), Bacteroidetes (1.95%–6.52%), and Actinobacteria (0.91%–4.81%). The relative abundance of Proteobacteria increased with prolonged storage time whereas those of the other phyla of bacteria slightly changed in the duration of cold-chain storage. We could infer that Proteobacteria played a major role in the spoilage of Spanish mackerel. These results were similar to the works studied by Cheng et al. [60] on mackerel during 4 °C refrigerated storage, which were based on high-throughput sequencing. It indicated that the little change in storage temperature has less effect for the SSO of mackerel.

Ten identified genera of bacteria, including Psychrobacter, Acinetobacter, Flavobacterium, Brochothrix, Carnobacterium, Vagococcus, Arthrobacter, Pseudomonas, Paeniglutamicibacter, and Shewanella, had more than 1% ratio for the six cold-chain Spanish mackerel samples (Figure 5B). The dominant bacteria in fresh Spanish mackerel belonged to Psychrobacter (52.76%). Pseudomonas gradually replaced Psychrobacter as the dominant bacteria just before the point of sensory rejection. The relative abundance of Pseudomonas changed from the initial 2.72% to 42.98%. On the contrary, the relative abundance of Psychrobacter decreased during cold-chain storage from 52.76% to 14.01%. Thus, we can infer that Pseudomonas was the SSO of Spanish mackerel and played an important role in the quality deterioration of cold-chain Spanish mackerel. Many reports described that nonfermenting Gram-negative bacteria were the main component of the spoilage organisms [61,62]. Pseudomonas is a nonfermenting Gram-negative bacteria and can cause an offensive odor by producing ammonia and trimethylamine N-oxide [63]. Therefore, techniques that can strategically inhibit the growth and metabolism of Pseudomonas will help to improve the storage quality and extend the shelf life of cold-chain Spanish mackerel.

## 4. Conclusions

In conclusion, we found that the SSOs might be helpful in providing a basis for the development of methods and tools to predict the shelf life of Spanish mackerel. The increased microbial activities (TVC and LAB), increased chemical activities (TVB-N and TBA), and decreased sensory activities (body surface, odor, fish gills, fish elasticity and eyes) with prolonged circulation and storage time during 0 °C cold-chain logistics proved that the cold-chain logistics could extend the shelf life of Spanish mackerel by approximately three days compared with traditional low-temperature storage methods (stored on ice at 3 °C refrigerator temperature). HTS was used to demonstrate the bacterial diversity and composition of cold-chain Spanish mackerel. The dominating phylum in the fresh and spoilage Spanish mackerels was Proteobacteria. The SSO genus was Pseudomonas. We deduced that we could extend the shelf life of cold-chain Spanish mackerel by inhibiting the growth of Pseudomonas. We could also use the number of Pseudomonas, the contents of TVB-N and TBA, and sensory characteristics to deduce the freshness of Spanish mackerel during the whole cold-chain in time because of the good correlation between sensory quality and chemical or microbial indicators. These results provided insights into the innovation of preservation technology and transportation methods for Spanish mackerel.

## Figures and Tables

**Figure 1 foods-09-00312-f001:**
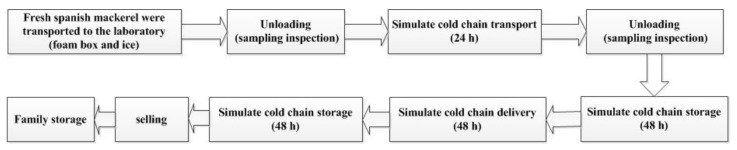
Fresh Spanish mackerel in simulated cold-chain logistics processes.

**Figure 2 foods-09-00312-f002:**
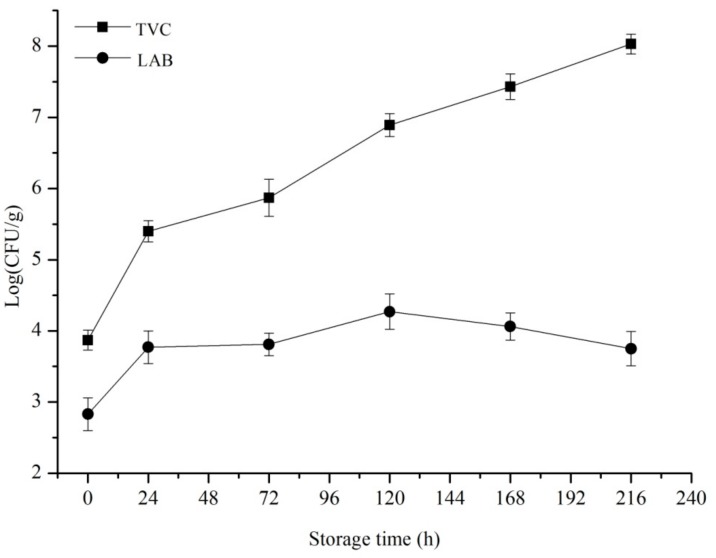
Changes in TVC (total viable count) (■) and LAB (lactic acid bacterial) (●) of Spanish mackerel at 0 °C cold chain storage. Data represent the mean ± SD values that were obtained from three samples.

**Figure 3 foods-09-00312-f003:**
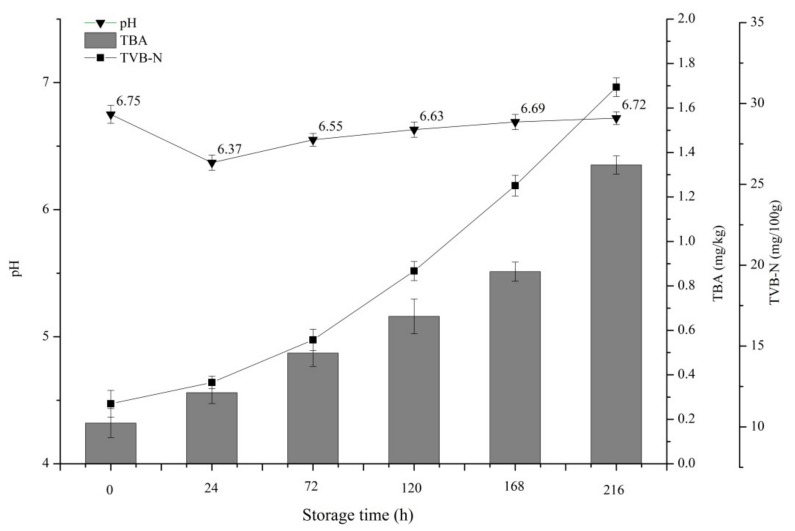
Changes in TVB-N, pH, and TBA of Spanish mackerel at 0 °C cold chain storage conditions. Data represent the mean ± SD values that were obtained from three samples. (■) represents TVB-N values, (▼) represents pH values, and (dash gray) represents TBA values. Abbreviations: TVB-N, total volatile base nitrogen; TBA, 2-thiobarbituric acid.

**Figure 4 foods-09-00312-f004:**
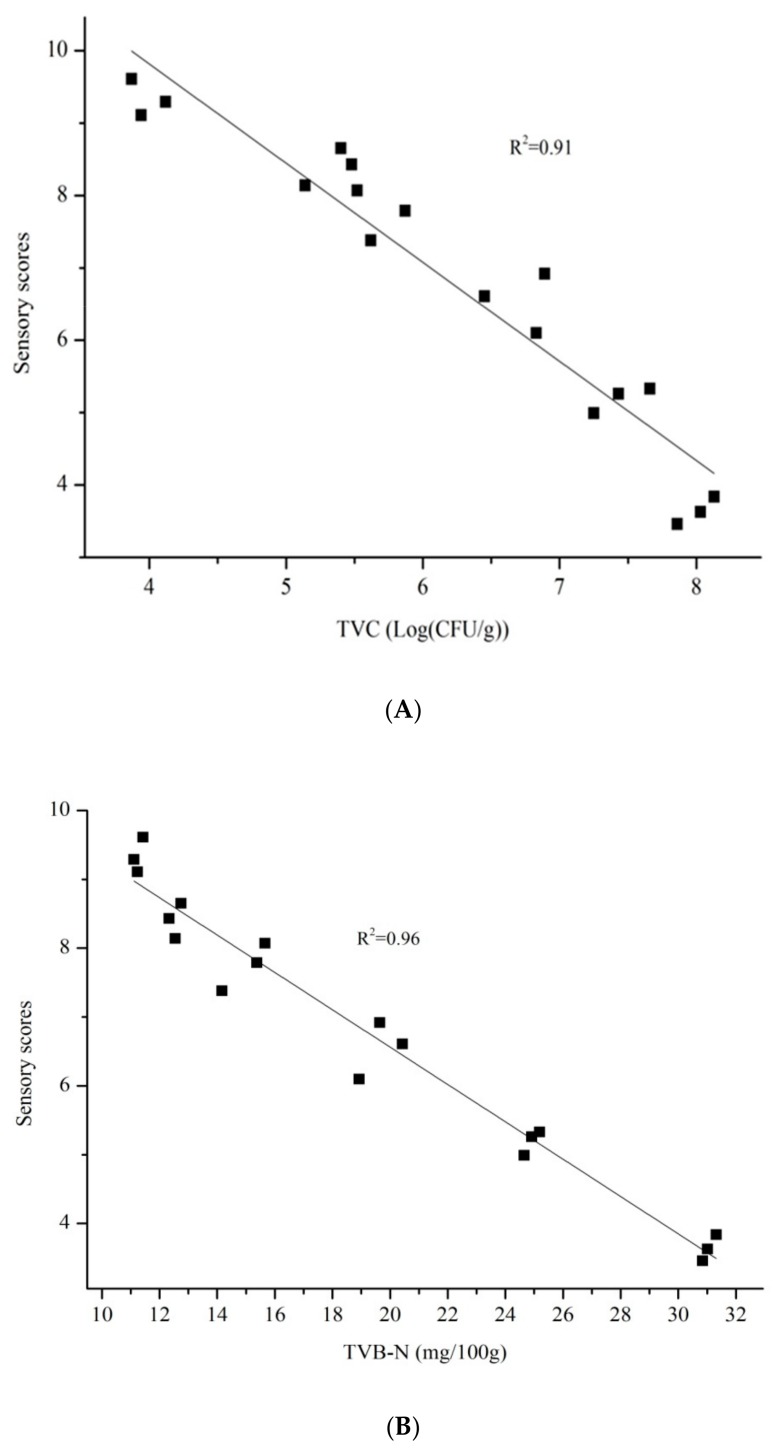
(**A**) Relationship between the overall assessment of sensory scores and TVC from the beginning of Spanish mackerel storage until the sensory rejection time. (**B**) Relationship between the overall assessment of sensory scores and TVB-N from the beginning of Spanish mackerel storage until the sensory rejection time. (**C**) Relationship between overall assessment of sensory scores and TBA from the beginning of Spanish mackerel storage until the sensory rejection time. Abbreviations—TVC, total viable count; TVB-N, total volatile base nitrogen; TBA, 2-thiobarbituric acid.

**Figure 5 foods-09-00312-f005:**
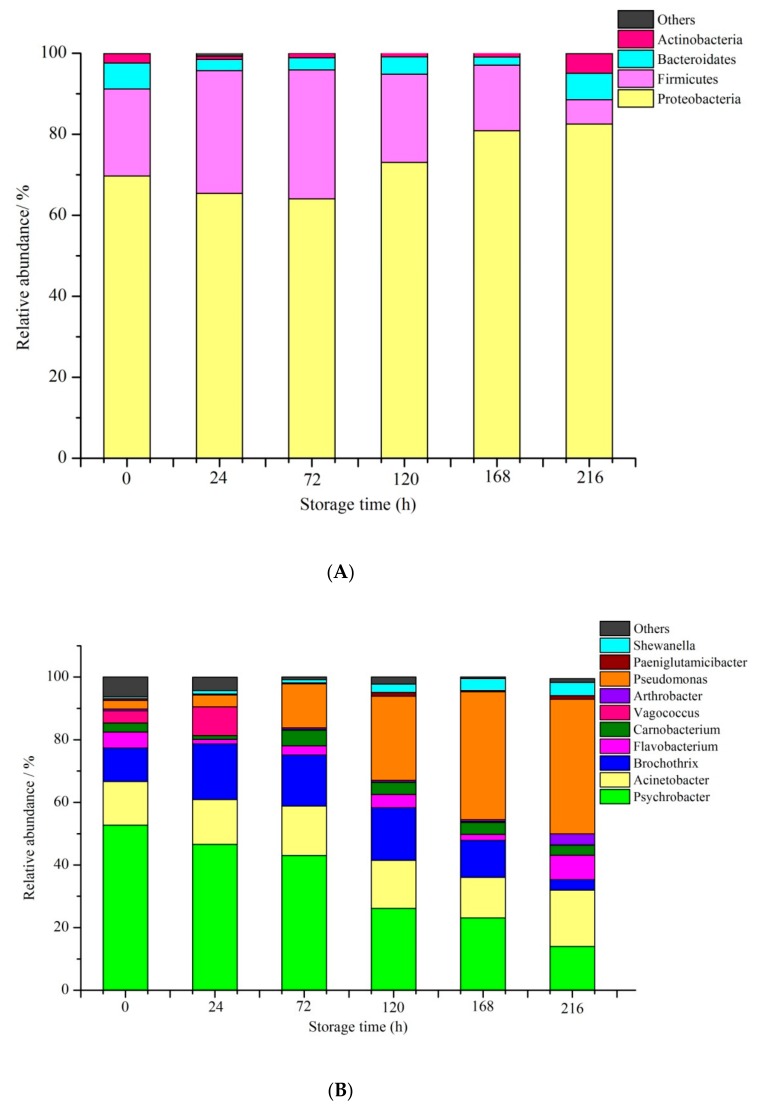
Relative abundance of the microorganisms in different Spanish mackerel samples—(**A**) bacterial composition at phylum level; (**B**) bacterial composition at genus level.

**Table 1 foods-09-00312-t001:** Sensory Evaluation Standard of Spanish Mackerel at 0 °C Cold-Chain Storage Conditions.

Score	Whole Fish				
	Body Surface	Odor	Fish Gills	Fish Elasticity	Eyes
9–10	Lustrous body surface intact without mucus	Odor inherent in fish	Bright red or burgundy with transparent mucus	Muscle is firm; tissue is tight and elastic	Full eyeball and clear cornea
7–8	Lustrous, slime transparent	Natural smell, no odor	Red; mucus is more transparent	Firm and elastic; the finger depressions recover quickly after pressing	Eyeball is flat and cornea is clear
5-6	Luster is a bit poor	Natural odors, slight odors	Reddish or dark red, mucus slightly cloudy	Muscles are softer and less elastic	Eyeball is flat or slightly sunkenand cornea is slightly cloudy
3–4	Surface of the body is dim with poor luster	Stinky	Mucus is cloudy and reddish	Poor elasticity; depression recovery after compression is slow	Eyeball is sunken and bleached; cornea is cloudy
1–2	Surface of the body is dull	Distinct odor of ammonia	Soil is yellow and the mucus is cloudy	Meat is loose	Eyeball is white; cornea is seriously turbid

**Table 2 foods-09-00312-t002:** Sensory Score of Spanish Mackerel at the 0 °C Cold Chain Storage Condition.

Index	Time (h)					
	0	24	72	120	168	216
Body surface	9.78 ± 0.17	8.81 ± 0.23	8.03 ± 0.12	6.98 ± 0.19	5.74 ± 0.24	4.48 ± 0.28
Odor	9.55 ± 0.14	8.58 ± 0.22	7.61 ± 0.24	6.82 ± 0.17	5.34 ± 0.45	3.55 ± 0.33
Fish gills	9.61 ± 0.15	8.52 ± 0.14	7.81 ± 0.27	6.86 ± 0.17	5.19 ± 0.25	3.51 ± 0.25
Fish elasticity	9.65 ± 0.15	8.70 ± 0.15	7.84 ± 0.14	6.62 ± 0.24	5.06 ± 0.15	3.40 ± 0.30
Eyes	9.45 ± 0.25	8.61 ± 0.17	7.66 ± 0.25	6.83 ± 0.15	4.96 ± 0.15	3.20 ± 0.13
Overall assessment	9.61 ± 0.11	8.65 ± 0.10	7.79 ± 0.015	6.82 ± 0.12	5.26 ± 0.27	3.63 ± 0.44

**Table 3 foods-09-00312-t003:** The Number of Effective Tags, Base Numbers, OTUs and Average Length of the Genome of Spanish Mackerel at Different Time Periods.

Groups	Effective Tags	Bases Number/bp	Average Length/bp	OTU Number
M0	41386	18587195	449.12	196
M24	46608	20971270	449.95	47
M72	42977	19314316	449.41	36
M120	51234	23051996	449.94	56
M168	62616	28039637	447.80	73
M216	55179	24609908	446.00	243

Notes—M0–M216 represent the respective number of hours Spanish mackerel was stored. OTU (Operational taxonomic units) is the same mark artificially set for a taxonomic unit (strain, species, genus, group, etc.) in phylogenetic or population genetics research for the convenience of analysis.

**Table 4 foods-09-00312-t004:** Alpha Diversity Indices and Coverage of Spanish Mackerel.

Groups	Shannon Index	Simpson Index	ACE Index	Chao1 Index	Coverage (%)
M0	2.39	0.15	174.84	120.21	0.9997
M24	2.34	0.18	123.40	108.00	0.9996
M72	1.92	0.29	109.21	84.11	0.9981
M120	1.95	0.25	196.67	137.27	0.9993
M168	2.47	0.13	328.33	182.10	0.9997
M216	2.03	0.22	241.54	138.15	0.9974

Notes—M0–M216 represent the respective number of hours Spanish mackerel was stored. ACE and Chao1, microbial richness estimators. Shannon and Simpson, microbial diversity estimators.

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
