# Peer review of "Quality Evaluation and Characterization of Specific Spoilage Organisms of Spanish Mackerel by High-Throughput Sequencing during 0 °C Cold Chain Logistics"

_foods, 2020, doi:10.3390/foods9030312_

Round 1

Reviewer 1 Report

The authors reported the “Quality evaluation and characterization of specific spoilage organisms of Spanish mackerel by high throughput sequencing during 0 °C cold chain logistics”, which, IMHO, it is a relevant study, but there are some important points that should be clarified before this work can be accepted for publication.

Line 208: Authors state that “Generally, fish are considered fresh and sub fresh when pH is 5.5-6.8 and 6.9-7.0, respectively, and fish are considered spoiled when pH exceeds 7.1 [ 52]. The authors must support better this information because “sub fresh” level seems too narrow (6.9-7.0) when compared with the limits for fresh fish (5.5-6.8). Related with this, the initial pH of Spanish mackerel (6.75 ± 0.03, line 197) is very close to the upper limit of the fresh interval. It is almost in the “sub fresh” level. Why? This does not seem to be consistent with the indication that the “freshly caught Spanish mackerel were collected from the Zhoushan fishing ground and transported to the laboratory in ice within 30 min (line 83). Authors should explain better this apparent inconsistency.

Line 211: Authors indicate that “the pH of spoiled Spanish mackerel was far from 7.1 when it became corrupted” and “speculate that pH was an inappropriate spoilage indicator for cold chain Spanish mackerel”. Authors should try to extrapolate the pH influence from similar studies in the literature.

Authors identified Pseudomonas as the main bacteria responsible for the fish degradation and indicate that “we could extend the shelf life of cold chain Spanish mackerel by inhibiting the growth of Pseudomonas”. How? This would be an interesting experiment to include in this report and justify why authors performed the HTS to identify the bacteria responsible for the fish degradation.

Authors claimed that “the cold chain logistics could extend the shelf life of Spanish mackerel by approximately three days compared with traditional low temperature storage method”. IMHO authors should have evaluated both storage methods in parallel so a direct comparison of the efficiency of both strategies could be assessed more clearly.

Line 63: Authors indicate that “several researchers have assessed the effects of cold chain logistics on the quality maintenance of grape [ 21], hairtail fish, and shrimp under certain conditions [ 22,23]. Results from these studies proved that cold chain logistics is more efficient than conventional storage for the preservation of agricultural products.” I must confess my scepticism regarding this preservation method because at 0ºC we are basically reaching the frozen point for water and such a low temperature can affect the organoleptic properties of the fish after cooked. This method can be more effective to preserve fish form spoilage, adding three more days to the logistic chain, but I would like to see a direct comparison on the effect of both procedures on the organoleptic properties of the fish.  

Minor points:

Line 166: The authors present the reference for fish freshness as CFU/g and then compare that value with the data they obtained in log CFU/g. Please use the same units to make the comparison more consistent and readily comparable.

Line 171: The TVC value reached 8.03 ± 0.14 log CFU/g when Spanish mackerel was stored for 216 h , which indicated that the Spanish mackerel sample had seriously deteriorated. So, this is 9 days, which means  than 8 days would be a safer estimation for consumption?

Fig. 3: pH representation in the figure a little confused. Please insert pH7 in the yy axis and eventually the values above the bars so we can appreciate more clearly the pH variation during the cold chain preservation of the Spanish mackerel.

Reviewer 2 Report

This manuscript uses several approaches to characterize the shelf life of Spanish mackerel during low temperature storage. Approaches used in the study are sound and properly executed, but produce results (correlation between acceptability and TVBN, Pseudomonas as primary spoilage organism) that are already well documented and accepted in diverse finfish. As such, justification for the study is lacking.

There are no references given to substantiate the claim that Spanish mackerel is an economically important fishery or that spoilage of this species is a particular problem in the current handling chain. Without such justification, the objectives of the study are unconvincing.

Materials and methods require significant additional detail for repeatability.

Line 56: The justification given for use of high throughput sequencing is invalid as this technique is generally not suitable to "fully examine the microbial species". Indeed, authors present results of this technique only to the genus level.

Line 71: identification of bacteria does not require sensory or chemical studies.

It is puzzling why authors did not choose to enumerate microorganisms specifically capable of growth at low temperatures. A psychrotrophic count would have been the most appropriate indicator of bacterial spoilage, but this is absent from the experimental design.

Line 87: there is an unnecessary "t"

Necessary details regarding the sterility of ice, coolers, and implements used for handling samples is not present.

Line 92: What type of thermometer was used? Model? K type, T type, variability?

Line 93: Where in the fish were probes placed? How was consistency in placement ensured?

Line 97: What modifications?

Figure 1: should "family storage" and "selling" be specified as simulated?

Line 111: Incubation time/temp for TVC?

Line 130: What constitutes an "experienced panel"? Were panelists trained? Were reference anchors given for the 10 point scale used? Was any verification of panel rating homogeneity undertaken?

Line 132: If overall acceptability is a calculated value then it should be absent from the prior list of attributes rated by panelists.

Line 175: Are any of these changes in population statistically significant?

Figure 2 caption needs explanation of abbreviations, indication of what error bars represent, number of samples, etc. Same comment for other figure legends.

Line 198: Are pH changes significant?

Line 205: Reduced acid production would not correlate to increasing pH, it would correlate to reduced rate of pH decrease.

Line 231: As compared to what?

Figure 5: The necessity of this figure is questionable. 

Line 304: Why would microbial consortia be assumed similar between oysters and mackerel? If they would not, this statement is superfluous.

Round 2

Reviewer 1 Report

The authors incorporated important information, discussions and corrections, and the original manuscript was clearly improved. Therefore, I now feel comfortable to recommend this report for publication. 

Reviewer 2 Report

Authors have addressed all suggestions from preliminary review sufficiently.